# Therapeutic Potential of Molecular Hydrogen in Metabolic Diseases from Bench to Bedside

**DOI:** 10.3390/ph16040541

**Published:** 2023-04-04

**Authors:** Fei Xie, Yifei Song, Yang Yi, Xue Jiang, Shiwen Ma, Chen Ma, Junyu Li, Ziyi Zhanghuang, Mengyu Liu, Pengxiang Zhao, Xuemei Ma

**Affiliations:** 1Faculty of Environment and Life, Beijing University of Technology, Beijing 100124, China; xiefei990815@bjut.edu.cn (F.X.);; 2Beijing Molecular Hydrogen Research Center, Beijing 100124, China

**Keywords:** molecular hydrogen (H_2_), metabolic diseases, oxidative stress, pre-clinical studies, clinical trials

## Abstract

Oxidative stress and chronic inflammation have been implicated in the pathophysiology of metabolic diseases, including diabetes mellitus (DM), metabolic syndrome (MS), fatty liver (FL), atherosclerosis (AS), and obesity. Molecular hydrogen (H_2_) has long been considered a physiologically inert gas. In the last two decades, accumulating evidence from pre-clinical and clinical studies has indicated that H_2_ may act as an antioxidant to exert therapeutic and preventive effects on various disorders, including metabolic diseases. However, the mechanisms underlying the action of H_2_ remain unclear. The purpose of this review was to (1) provide an overview of the current research on the potential effects of H_2_ on metabolic diseases; (2) discuss the possible mechanisms underlying these effects, including the canonical anti-oxidative, anti-inflammatory, and anti-apoptotic effects, as well as suppression of ER stress, activation of autophagy, improvement of mitochondrial function, regulation of gut microbiota, and other possible mechanisms. The potential target molecules of H_2_ will also be discussed. With more high-quality clinical trials and in-depth mechanism research, it is believed that H_2_ will eventually be applied to clinical practice in the future, to benefit more patients with metabolic disease.

## 1. Introduction

Molecular hydrogen (H_2_) is a colorless, tasteless, and odorless gas, which has long been considered physiologically inert. Its special properties, including small size, low mass, neutral charge, nonpolarity, and high rate of diffusion, enable it to easily penetrate cellular biomembranes and rapidly diffuse into the cytosol, mitochondria, nucleus, and other organelles [1], which stimulates the scientist to explore its potential biological function. In 1975, Dole et al. first reported that hyperbaric H_2_ therapy (absolute pressure: 8 atm, 2.5% O_2_ and 97.5% H_2_) for 2 weeks could induce significant tumor regression in mice with squamous cell carcinoma [2]. In 2001, Gharib et al. provided evidence that hyperbaric treatment (absolute pressure: 8 atm) for 2 weeks could significantly alleviate parasite-induced liver inflammation in mice [3]. However, due to the high demand for hyperbaric H_2_ therapy and the potential explosion hazards of H_2_, these studies did not attract much attention. The turnaround occurred in 2007, when Ohta’s group reported that inhalation of low concentration of H_2_ (2%) at normal atmospheric pressure could markedly attenuate cerebral ischemia–reperfusion injury in rats by selectively scavenging hydroxyl radicals and peroxynitrite [4]. Subsequently, further studies have been conducted to explore the potential biological effects of H_2_ in a wide range of disease models, including metabolic diseases, neurodegeneration, mitochondrial diseases, inflammation, and cancer [5]. Among these studies, H_2_ has been administered by multiple routes, including oral intake of hydrogen-rich water (HRW), injection of hydrogen-rich saline (HRS), inhalation of H_2_ gas, HRW baths, and hydrogen-producing materials. Although the therapeutic and preventive potentials of H_2_ have been extensively reported in both pre-clinical and clinical studies, inconsistencies of intervention effects across studies may exist due to the influence of various factors, including the H_2_ concentration, treatment duration, and the time point of H_2_ intervention. In addition, the mechanism underlying the action of H_2_ has not been clearly elucidated as yet.

Metabolic diseases, including diabetes mellitus (DM), metabolic syndrome (MS), fatty liver (FL), atherosclerosis (AS), and obesity, are characterized by dyslipidemia, insulin resistance, defective insulin secretion, glucose intolerance, and chronic inflammation. Oxidative stress has been implicated in the pathophysiology of metabolic diseases [6]. A considerable amount of literature has been published on the potential beneficial effects of H_2_ intervention in metabolic diseases (Figure 1), including improving glucose and lipid metabolism, alleviating pancreatic beta-cell damage, attenuating hepatic steatosis, promoting atherosclerotic plaque stabilization, etc. In this review, we aim to summarize the current knowledge on the protective effects of H_2_ against metabolic diseases and the potential mechanisms involved.

## 2. Protective Effects of H_2_ against Metabolic Diseases

### 2.1. Diabetes Mellitus

DM is a chronic metabolic disorder characterized by chronic hyperglycemia, resulting from defects in insulin secretion and/or insulin function. DM is mainly categorized into two sub-types: type 1 DM (T1DM) and type 2 DM (T2DM). T1DM is an autoimmune disorder that attacks the insulin-producing pancreatic beta cells with T cells, resulting in an absolute insulin deficiency [7]. As the most prevalent form of DM, T2DM, which is also known as non-insulin-dependent diabetes, is mainly caused by impaired insulin secretion by pancreatic beta cells and resistance of insulin-sensitive tissues to insulin action, resulting in a relative lack of insulin [8]. DM can cause multiple complications, including macrovascular complications (such as coronary heart disease, stroke, and peripheral vascular disease) and microvascular complications (such as end-stage renal disease (ESRD), retinopathy, and neuropathy) [9]. Although considerable research has been devoted to the development of diabetes medications, their efficacies are often hampered by various side-effects, thus necessitating the search for novel therapeutic agents for the treatment of DM.

A growing body of evidence has demonstrated that excessive reactive oxygen species (ROS) generation, largely due to hyperglycemia, causes oxidative stress, which further promotes the development and progression of DM and its complications [10]. Pancreatic beta cells are highly vulnerable to oxidative stress, due to their high endogenous production of ROS and low levels of antioxidant enzyme expression [11]. The destruction of beta cells is a pathological component of T1DM and T2DM. As a novel antioxidant agent, the potential therapeutic effects of molecular hydrogen on DM and its complications have been extensively investigated in both pre-clinical and clinical studies, as summarized in Table 1. The attenuation of oxidative stress and enhancement of an anti-oxidant response, as evidenced by decreased malondialdehyde (MDA) levels and increased glutathione (GSH) levels and superoxide dismutase (SOD) and catalase (CAT) activities, is at least partially responsible for the antidiabetic effect of H_2_ on DM [12,13,14,15,16,17].

To investigate the effect of molecular hydrogen on T2DM, two types of animal models have been used in pre-clinical research: (1) insulin resistance obesity and diabetes models, including Lepr^*db*/*db*^ mice and HFD-induced obesity mice or rats. These animals exhibited a significant increase in body weight, and FBG, insulin, TG, and TC levels. (2) HFD + STZ-induced mice or rat models. As HFD induces insulin resistance and STZ causes partial beta cell destruction, these models closely mimic natural disease progression (from insulin resistance to beta cell dysfunction), as well as the metabolic characteristics of T2DM [21]. Based on these models, hydrogen was administered to animals via four routes, including drinking HRW, oral gavage of HRS, injection of HRS, and injection of hydrogen gas, for different durations (from 2 weeks to 3 months). Until now, no research has been conducted to compare the effect of hydrogen delivery methods, treatment duration, and hydrogen dosage on the FBG-lowing effect in detail, although several studies have provided some clues. Different hydrogen intervention methods may present different change patterns in FBG levels; for example, the FBG levels dropped quickly during the first two weeks of subcutaneous injection of hydrogen gas, followed by a slow reduction during the next two weeks [13], while a significant decrease in FBG levels did not occur until the third week of HRW intake [12]. Low-concentration HRW has been shown to have a smaller FBG-lowering effect to high-concentration HRW [17]; however, the dosage effects of H_2_ on DM need to be further investigated. It is worth noting that the FBG-lowering effect with higher initial FBG levels prior to intervention seems to be more apparent, especially in HFD + STZ-induced DM models.

Insulin resistance and beta-cell dysfunction play an important role in the pathogenesis and evolution of T2DM [22]. The anti-T2DM effect of H_2_ has been associated with improved insulin sensitivity, evidenced by increased insulin sensitivity index (ISI), decreased insulin resistance index (IRI) and homeostasis model assessment-insulin resistance (HOMA-IR) index, and improved glucose and insulin tolerance [13,14,16]. The insulin-sensitizing effect of H_2_ is even slightly better than pioglitazone [14]. H_2_-induced upregulation of insulin and insulin receptors in both the adipose and skeletal muscle tissues may partially contribute to the improved insulin sensitivity [15]. H_2_ also showed a protective effect with respect to STZ-induced beta-cell damage, although the evidence was not definitive. A histopathological examination revealed a significant attenuation of STZ-induced pancreatic islet morphological changes after H_2_ intervention [16]. STZ-induced decreases in insulin levels could also be ameliorated by H_2_ [15], which indirectly confirmed the protective role of H_2_ in beta-cell dysfunction. It is worth pointing out that in some obesity and HFD + STZ-induced animals, hyperinsulinemia secondary to hyperglycemia and insulin resistance could also be restored by H_2_ [13,14,17].

The modulating role on metabolism may also contribute to the anti-T2DM effect of H_2_. Hydrogen-induced alterations in hepatic glucose and lipid metabolism have been reported in several studies. Kamimura et al. [17] found that drinking HRW may stimulate energy metabolism, through upregulation of hepatic fibroblast growth factor 21 (FGF21) expression. It has been established that the liver-derived metabolic hormone FGF21 plays a key role in the regulation of energy homeostasis [23]. The induction of FGF21 expression may be responsible for the decrease in plasma triglyceride and glucose induced by hydrogen administration. It has been reported that increased hepatic FGF21 expression can also be induced by dietary manipulation, such as fasting or ketogenic diet [23]. Consistently, diet restriction and HRW consumption showed a similar downregulatory effect on plasma glucose, insulin, and triglycerides levels. Hydrogen administration has also been found to promote the synthesis of hepatic glycogen, whilst improving the utilization of glucose in the liver [16]. In addition, a reduction of hepatic fat accumulation can also be observed, even with short-term hydrogen interventions (1–2 weeks), suggesting a significant effect of hydrogen on hepatic lipid metabolism [12,17]. The reduced serum levels of TG, TC, and LDL-c, and increased HDL-c levels, also support the regulatory effect of hydrogen on lipid metabolism [12,13,14,16,17].

Unlike T2DM, research on the effect of H_2_ on T1DM is very limited. Amitani et al. [18] established an STZ-induced type 1 diabetic mice model and found that chronic intraperitoneal and oral administration of H_2_ significantly improved glycemic control. In fact, as early as 2006, Kim et al. [19] provided the first in vivo evidence of the antidiabetic effect of electrolyzed reduced water (ERW) in both T1DM and T2DM, as evidenced by decreased FBG levels and improved glucose tolerance. ERW, also called “alkaline ionized water”, is characterized by alkaline pH, low dissolved oxygen gas concentration, high dissolved hydrogen gas concentration, and a negative oxidation–reduction potential (ORP) [24]. The anti-T1DM effect of ERW has also been observed in an alloxan-induced insulin-dependent diabetic mice model [20]. Increasing evidence demonstrates that the dissolved hydrogen in ERW is exclusively responsible for any observed therapeutic effects [24]. From current research, the inhibition of oxidative stress and beta-cell apoptosis and promotion of glucose uptake into skeletal muscle and insulin sensitivity in peripheral tissues may contribute to the anti-T1DM effects of H_2_ [18,19,20].

The protective effects of H_2_ against the diabetic complications, including peripheral neuropathy [25,26,27,28,29], retinopathy [30,31], cardiomyopathy [32,33], wound healing [34,35,36], bone loss [37], and erectile dysfunction [38], have been extensively reported. STZ-induced mice or rats models were used in the above in vivo studies for diabetic complications. In addition to conventional hydrogen delivery routes, some novel administration methods, such as photo-driven nanoreactor, microneedle patch, and algae–bacteria gel patch, were used, which allowed in situ continuous hydrogen production to treat chronic diabetic wounds. The inhibition of oxidative stress, inflammation, endoplasmic reticulum stress, apoptosis, and pyroptosis, as well as parthanatos (a particular type of programmed cell death that is different from the common apoptosis, necrosis, and autophagy) and activation of the mitochondrial ATP-sensitive potassium (Mito-K-ATP) pathway may be involved in the protective effects of H_2_ on diabetic complications [25,26,27,28,29,30,31,32,33,34,35,36,37,38].

To date, three clinical trials have been conducted to explore the potentials of H_2_ in the treatment of DM (Table 2). The first clinical trial was conducted in 2008 by Kajiyama et al. [39], showing a potential role of HRW in the prevention of T2DM and insulin resistance, as evidenced by normalized glucose tolerance in four of six patients with impaired glucose tolerance (IGT); however, no changes were observed in the levels of FBG and insulin. HRW intake also induced decreased levels of modified LDL, especially small dense LDL and urinary 8-isoprostanes; however, additional research is needed to clarify the regulation of these biomarkers. The other two clinical trials were both aimed at exploring the potential effects of ERW on T2DM. One of the two clinical studies was a multicenter, prospective, double-blind, randomized controlled trial led by Tohoku University [40]. Drinking ERW for three months induced a significant reduction in serum lactate levels, but no changes were observed in HOMA-IR and levels of FBG and fasting plasma insulin (FPI). Further in-depth analysis revealed that in the HOMA-IR ≥1.73 group, the rate of change in lactate levels was positively correlated with the rate of change in HOMA-IR, FBG and FPI levels; in the FBG ≥6.1 mM group, the FPI levels dropped significantly and FBG levels reduced with a marginal significance (*p* = 0.067). These findings indicated that patients with more serious conditions, such as more insulin resistant or higher FBG levels, may obtain more benefits from hydrogen treatment, which is consistent with pre-clinical studies. Another clinical study showed that oral intake ERW for 12 days decreased total cholesterol and improved other lipid profiles in patients with T2DM accompanied by dyslipidemia [41]. An FBG-lowering effect was also observed in the ERW group compared with the controls, although the difference was not significant. In conclusion, potential beneficial effects of H_2_ for T2DM were observed in recent clinical studies. However, due to the small number of participants and short periods of intervention, the therapeutic effect of H_2_ on T2DM is not obvious. Further high-quality clinical trials, such as large-scale, multicenter, prospective, double-blind, randomized control clinical trials of H_2_ for the prevention and treatment of T2DM, T1DM, and DM-related complications are needed in the future.

### 2.2. Metabolic Syndrome

MS is typically characterized by the simultaneous occurrence of at least three of the following medical conditions: obesity, hyperglycemia, hypertension or dyslipidemia, which predispose individuals to cardiovascular disease and T2DM [52]. The pre-clinical studies conducted by Hashimoto et al. [53] showed that intake of HRW for 16 weeks had no effect on TG, TC, FBG, and blood pressure in SHR.Cg-Leprcp/NDmcr (SHR-cp) rats; however, HRW could exert beneficial effects against renal abnormalities, as evidenced by preventing glomerulosclerosis and ameliorating creatinine clearance. The enhancement in antioxidative activity and inhibition ROS production in the kidneys may be responsible for the effect of H_2_ on MS-related renal dysfunction. Zhao et al. [54] provided indirect evidence that oral administration of L-arabinose, a naturally occurring plant pentose, can induce hydrogen gas production through gut fermentation and alleviate HFD-induced MS, including gain in body weight and fat, impaired insulin sensitivity, liver steatosis, dyslipidemia, and elevated inflammatory cytokines, indicating that increasing endogenous hydrogen gas release through modulating gut microbial composition by administration of prebiotic nutrients or hydrogen-producing probiotics might be a promising approach for the treatment of MS. In conclusion, these pre-clinical findings revealed the potential beneficial effects of H_2_ on MS (Table 3).

To further characterize the effects of H_2_ on patient with MS, three clinical studies have been conducted to date (Table 2). Nakao et al. first performed an open label pilot study on 20 subjects with potential MS and demonstrated that drinking HRW (~1 mmol H_2_/day) for 8 weeks could increase HDL-c levels by 8% and decrease the total cholesterol-to-HDL ratio by 13% [42]. These effects may have been caused by the antioxidant effect of H_2_, as evidenced by SOD levels increased by 39% and decreased TBARs by 43%. Similarly, song et al. reported that HRW intake (0.5 mmol H_2_/day) for 10 weeks in patients with potential MS decreased TC and LDL-C levels, improved HDL function, and alleviated oxidative stress and inflammation [43]. Recently, LeBaron et al. carried out a randomized, double-blinded, placebo-controlled trial in patients with MS [44]. Compared with the previous two studies, this trial included more subjects (60 MS patients), and, importantly, higher concentration HRW (>5.5 mmol H_2_/day) was administered for a longer duration (24 weeks). Consistently with previous studies, HRW intake decreased the levels of TG and LDL-c, TC/HDL-c ratio, inflammation, and lipid peroxidation. However, this study also found some different effects from previous studies: (1) The body mass index (BMI), waist–hip circumference, and resting heart rate were significantly decreased, all these changes were absent in the previous two studies. (2) FBG levels decreased after a 24-week HRW intervention with an accompanying 12% reduction in HbA1C, while this glucose-lowering effect was not observed in previous studies. (3) The levels of HDL-c were significantly decreased, whereas in the open-label study, there was an increase in HDL levels. A reduction in HDL-c was also observed in T2DM rats after HRW intake [12]. This was probably due to the dramatic reduction in TC levels induced by H_2_, resulting in a lower TC/HDL-c ratio, which is associated with a decreased risk of cardiovascular disease. The longer duration with high-concentration of HRW may have been responsible for these different effects. In conclusion, HRW administration, especially long-term treatment with a high concentration of HRW, may be a promising adjuvant therapy to decrease the features of MS.

### 2.3. Fatty Liver

FL is characterized by a wide histological spectrum, ranging from steatosis, steatohepatitis (SH), to fibrosis/cirrhosis, and has traditionally been divided into alcoholic liver disease (ALD) and non-alcoholic fatty liver disease (NAFLD) [75]. ALD is caused by chronic and excessive alcohol consumption, while NAFLD is defined by evidence of FL without alcohol abuse. NAFLD has been considered a continuum from obesity to MS and DM [76]. Due to the close association of NAFLD with metabolic risks, it has recently been proposed to change its name to metabolic (dysfunction)-associated fatty liver disease (MAFLD) [77]. Oxidative stress has been considered to play a key role in the pathogenesis of both ALD and NAFLD [75]. The beneficial effects of H_2_ on both ALD and NAFLD were reported in previous pre-clinical studies. Lin et al. investigated the potential effect of HRW on ALD in chronic-binge ethanol-fed mice model [56]. The results showed that HRW protected against early-stage chronic EtOH-induced liver injury, possibly by reversing EtOH-induced anorexia via upregulation of acyl ghrelin expression, and suppressing inflammation and oxidative stress, which indicates the potential of HRW for prevention and clinical complementary treatment of ALD. Compared with ALD, more detailed and in-depth studies exploring the effects of H_2_ on NAFLD have been reported. As shown in Table 3, the evidence for H_2_-induced improvement of NAFLD includes (1) alleviation of hepatic steatosis, as evidenced by reduced hepatic lipid contents, hepatocellular ballooning degeneration, liver weight, and plasma ALT and AST activities; (2) reversion of steatohepatitis and fibrosis, supported by inhibition of hepatic inflammatory cell infiltration and hepatic stellate cell (HSC) activation, reduced necrotic areas and Sirius red-positive area in liver histopathology, downregulation of pro-inflammatory cytokines, and upregulation of anti-inflammatory cytokines; (3) attenuation of dyslipidemia, as evidenced by reduced body weight gain, BMI, WAT mass, and serum TC and TG; (4) improvement of glycemic control, as demonstrated by decreased FBG and fasting insulin (FINS) levels, and improved insulin sensitivity and glucose tolerance. In addition, it has also been found that H_2_ could ameliorate ischemia reperfusion injury in FL and prevent hepatocarcinogenesis [59,63]. Some of these studies investigated the dose effect of H_2_ and showed that higher doses of H_2_ administration demonstrated better therapeutic effects in most cases [60,62], even though a low dose of H_2_ showed better effects for some indices (e.g., plasma ALT and AST levels) [60]. Moreover, the study by Li et al. demonstrated that a longer duration of H_2_ intervention exhibited more obvious improvements [61]. Consistently, prolonged hydrogen release from a nanocapsule exerted more profound effects [62]. These studies also indicated that antioxidant, anti-inflammatory, and anti-apoptotic effects may be involved in the mechanism of action of H_2_ in FL (Table 3).

The currently published clinical studies mainly focused on the effects of H_2_ on NALFD (Table 2). A preliminary clinical study involving 12 overweight outpatients with NALFD was conducted by Korovljev et al. in 2019 [45]. The results showed that drinking HRW (3 mmol/day) significantly reduced liver fat accumulation and serum AST levels; however, there was no significant effect on the body weight, BMI, body composition, lipid profiles, and glucose levels. Recently, the effects of HRW were investigated in another clinical trial with 30 subjects [46]. HRW intervention was performed with a higher dose (>6 mmol/day) and longer duration (8 weeks), resulting in a favorable trend in body weight, BMI, SBP (reduced by ~2 mmHg), TG/HDL ratio, and LDH; however, most of the observed trends were not statistically significant. The lack of statistically significant changes may be due to either the values already being in homeostasis/range or a shorter duration of H_2_ intervention. Another recent study investigated the effect of 13-week hydrogen/oxygen (66% H_2_, 33% O_2_) inhalation on 43 subjects with NALFD [47]. Hydrogen/oxygen inhalation showed no effect on liver steatosis in mild cases; however, the liver fat content measured by ultrasound and CT scans was significantly improved in moderate–severe cases. Hydrogen/oxygen inhalation also significantly decreased LDL-c, AST, ALT, and biomarkers of oxidative stress and inflammation. In conclusion, the present studies have shown beneficial effects of H_2_ on NAFLD, especially in moderate–severe cases, a while higher dose of H_2_ intervention for a longer period had a tendency to be more effective.

### 2.4. Atherosclerosis

AS is a chronic progressive disease triggered mainly by abundant accumulation of apoB-containing lipoproteins and vascular inflammation in medium-sized to large arteries and that eventually leads to compromised blood flow [78]. One possible pathological mechanism underlying atherogenesis is that chronic stress causes endothelial injury, inducing the entry of monocytes into the subendothelium and their subsequent differentiation into macrophages, promoting foam cell formation and resulting in the formation of atherosclerotic plaque [79]. The potential effects of H_2_ on AS were investigated in apoE or LDL receptor (LDLR)-deficient mice. As demonstrated in Table 3, on the one hand, H_2_ intervention could downregulate apoB protein levels in plasma and in the liver, alleviate serum lipid oxidation, improve HDL functionality, and reduce inflammatory processes in the arterial wall, which may contribute to the inhibition of the initiation and development of atherosclerosis. On the other hand, H_2_ could increase plaque collagen content; reduce macrophage infiltration, lipid deposition, vascular smooth muscle cell (VSMC) content, cell apoptosis, and MMP-9 expression in atherosclerotic plaques; and alleviate the formation and apoptosis of macrophage-derived foam cells, which resulted in plaque stabilization. Higher doses of hydrogen intervention showed better anti-AS effects [66,67,69]. These studies indicated the potential role of H_2_ in preventing and treating atherosclerotic cardiovascular and cerebrovascular disease. The suppression of oxidative stress, inflammation, apoptosis, and ER stress, and activation of autophagy may be involved in the anti-AS effects of H_2_.

To date, no published clinical trials have directly revealed the effect of H_2_ on patients with AS; however, several clinical studies have provided evidence that H_2_ may exert inhibitory effects on the initiation and progression of AS, such as improving vascular endothelial function, and preventing and relieving peripheral arterial disease (PAD) (Table 2). Sakai et al. investigated the potential effects of high-concentration (3.5 mM) HRW on vascular endothelial function in 34 health subjects [48]. The endothelial function was assessed by measuring the flow-mediated dilation (FMD) of the brachial artery (BA) 30 min after drinking HRW. Compared with the placebo group, the relative ratio to the baseline in the changes of FMD after consumption of HRW was significantly improved, indicating the protective potential of H_2_ toward the endothelium. The improved endothelial function induced by H_2_ was corroborated by comparable observations in a recent clinical study from the same group [49]. The peripheral endothelial function in the small arteries of the fingers was evaluated using another non-invasive approach, reactive hyperemia-peripheral arterial tonometry (RH-PAT), in 68 healthy subjects. A significant improvement in natural logarithmic transformation of the reactive hyperemia index (Ln-RHI) was observed 24 h after the first consumption and continuous consumption of HRW for 2 weeks. To investigate the influence of Ln_RHI values before HRW intake on the effects of H_2_, the subjects were then divided into low and high Ln-RHI groups, for further analysis. Compared with the placebo group, daily intake of HRW for 2 weeks significantly improved the Ln_RHI value in the low Ln_RHI group, while no significant improvement was observed in the high Ln_RHI group, indicating that patients with higher risk for vascular events may benefit more from H_2_ interventions. Since endothelial dysfunction is an early event of AS, these results indicated the inhibitory effects of H_2_ on the initiation of AS. In conclusion, the present studies have provided preliminary evidence of the potential benefits of H_2_ for AS-related disease, and further clinical studies should be performed to confirm the anti-AS role of H_2_.

### 2.5. Obesity

Obesity is characterized by excessive body fat accumulation and chronic low-grade inflammation, and is often associated with increased risk of hypertension, T2DM, dyslipidemia, NAFLD, and cardiovascular disease (CVD) [80]. Many factors are involved in the pathogenesis of obesity, such as genetics, viruses, insulin resistance, inflammation, gut microbiome, circadian rhythms, and hormones [80]. Anti-obesity effects have been extensively reported in pre-clinical studies. These studies provided evidence that H_2_ could suppress weight gain and fat accumulation in the liver, alleviate plasma dyslipidemia, improve glycemic control, and extended the average of lifespan of HFD-induced-obesity mice (Table 3). Research by Masuda et al. showed that a short H_2_ intervention (2 weeks) had no effect on the blood sugar level in HFD-induced-obesity mice but changed the anti-metabolic phenotype of both WAT and BAT adipocytes, from hypertrophic to hyperplastic [73]. The findings of H_2_-induced rapid regulation of lipid metabolism in this study, along with evidence from other studies, such as the reduction in hepatic fat accumulation and the promotion of fatty acid metabolic gene expression in the liver found in obese mice after 1–2 weeks of HRW intake [17,71], indicated that H_2_ could restore lipid metabolism toward energy consumption more favorably than glucose metabolism. In addition to the suppression of oxidative stress and inflammation, the regulation of gut microbiota [72], the alleviation of oxidized phospholipid (OxPLs) accumulation by enhancing the anti-oxidative capacity of HDL [74], as well as the acceleration of BAT activation [73] may be involved in the anti-obesity effects of H_2_.

To date, the clinical research published on the anti-obesity effects of H_2_ has been very limited (Table 2). Korovljev et al. evaluated the effects of H_2_ intervention on body composition, hormonal status, and mitochondrial function in ten middle-aged overweight women [50]. Orally administration of hydrogen-generating minerals for 4 weeks resulted in a significant reduction in body fat, arm fat index, serum TG, and insulin levels, accompanied by a strong increasing trend of ghrelin secretion. However, H_2_ administration had no significant effect on BW, serum glucose, and other lipid parameters, which may have been caused by the relatively short treatment duration. It has been reported that fasting blood lactate accumulation reflects mitochondrial dysfunction, which in turn influences metabolic disease risk [81]. It is noteworthy that after 4 weeks of H_2_ intervention, the blood lactate levels were significantly lower than the placebo group (*p* = 0.01), indicating that an improvement of mitochondrial function may be involved in the anti-obesity effect of H_2_. Asada et al. also provided evidence that taking a 10 min warm (41 °C) HRW bath every day for 1–6 month could decrease the visceral fat accumulation and blood LDL-c levels in some subjects [51]. Although the current clinical studies have provided some evidence of the anti-obesity effects of H_2_, the reliability of the evidence is limited, due to the small number of subjects. Further large-scale trails with long-term H_2_ interventions should be performed, to investigate the anti-obesity effects of H_2_ in detail.

In conclusion, the results from current pre-clinical and clinical studies indicated the potential beneficial effects of H_2_ on metabolic diseases. The main effects of H_2_ on various tissues in metabolic diseases are illustrated in Figure 2.

## 3. Possible Mechanisms of H_2_ Action on Metabolic Diseases

The canonical mechanisms of H_2_ action, including anti-oxidative, anti-inflammatory, and anti-apoptotic effects, have been extensively reported in various diseases. In addition to these canonical effects, suppression of ER stress, and activation of autophagy, preservation of mitochondrial function, and regulation of gut microbiota may also be involved in the action of H_2_ on metabolic diseases. The possible mechanisms underlying the effects of H_2_ on metabolic diseases are illustrated in Figure 3.

### 3.1. Anti-Oxidative Effects of H_2_

The anti-oxidative effect of H_2_ has been confirmed in numerous disease models of metabolic diseases. H_2_ may exert its antioxidant activity in three ways, including directly scavenging free radicals, inhibiting the ROS generation, and enhancing antioxidant enzyme activity. Ohta’s group first proposed that H_2_ may play an antioxidant role by selectively scavenging hydroxyl radicals and peroxynitrite [4]; however, the increasing evidence of H_2_-induced biological effects cannot be fully explained by its radical scavenging properties [82]. Based on the theory of metal-free hydrogen activation by Lewis acid and base, known as the frustrated Lewis pair (FLP) mechanism, Ishibashi proposed that H_2_ could be activated in the Q-chamber of mitochondria, providing electrons and protons to the ubiquinone (UQ) species, including highly-reactive semiquinone intermediates, preventing the premature leakage of electrons from the ETC and leading to the suppression of ROS generation [82]. Ishihara et al. provided in vitro evidence that H_2_ mainly suppress superoxide generation in complex I [83]. In addition, Ma’s group demonstrated the existence of H_2_-evolving activity in eukaryotic mitochondria, which is closely related to complex I and could be promoted by hypoxia [84]. They proposed that hypoxia-induced oxidative stress could be inhibited by H_2_ production. In this process, hypoxia induced mitochondrial succinate accumulation and reverse electron transfer, leading to a reduction in the UQ pool, and protons then have the chance to compete with UQ for electrons, which eventually results in the release of H_2_. This evidence indicated that eukaryotic mitochondria, especially the complex I, may partially retain the activity of hydrogenase, allowing the conversion of dihydrogen into protons and electrons, and the generation of dihydrogen, which may play an important role in the regulation of ROS generation. In addition to directly reducing ROS, H_2_ could also indirectly suppress oxidative stress by increasing antioxidant capacity via regulating anti-oxidative signal transduction. H_2_ has been reported to activate the nuclear factor erythroid 2-related factor 2 (Nrf2)-mediated anti-oxidative pathway, leading to an increased expression of downstream target enzymes, including heme oxygenase-1 (HO-1), catalase (CAT), SOD, and GSH-Px [29,34,67]. Although H_2_ has been reported to ameliorate lipid accumulation by regulating the miR-136/MEG3/Nrf2 pathway in NAFLD [58], it is unclear whether H_2_ could also regulate Nrf2 activity through the canonical Keap1-Nrf2-ARE pathway or other non-canonical pathways in metabolic diseases [85].

In addition, previous studies also provided further evidence to explain how H_2_ exerts protective effects against metabolic diseases through antioxidation. Ohta’s group performed a microarray analysis to examine the changes in hepatic gene expression at different time points of H_2_ treatment in obesity mice [71]. Combined with their previous findings, they speculated that H_2_ reduced hydroxyl radicals, resulting in a decrease of peroxides and their end products, including 4-HNE, and thus activating PGC-1α through suppression of Akt/FoxO1 phosphorylation, followed by stimulation of the PPARα pathway, upregulation of FGF21 gene expression, and finally enhancing the fatty acid metabolism. PPARγ, another member of the PPAR nuclear receptor superfamily, was also upregulated after H_2_ treatment, which may synergistically improve glucose and lipid metabolism with PPARα [57]. In conclusion, H_2_ could suppress oxidative stress via directly scavenging free radicals, inhibiting the ROS generation, or increasing the cellular antioxidant capacity, leading to the alleviation of oxidative damage in various organs, including the liver and pancreas, as well as the improvement of glucose and lipid metabolism.

### 3.2. Anti-Inflammatory Effects of H_2_

The occurrence and development of metabolic diseases are closely associated with an uncontrolled immune response. Whereas in T1DM, pancreatic beta cells are destroyed by an autoimmune response, the onset of T2DM, MS, FL, AS, and obesity are accompanied by chronic low-grade inflammation. It has been extensively reported that H_2_ could alleviate inflammation in metabolic diseases through downregulation of pro-inflammatory cytokines, and upregulation of anti-inflammatory cytokines. The anti-inflammatory effects of H_2_ are always paralleled by its antioxidant effects. HO-1 is an important mediator of the cross-talk between anti-inflammatory and anti-oxidative pathways. It has been reported that H_2_ could inhibit the lipopolysaccharide (LPS)-induced production of TNF-α and other inflammatory cytokines via an HO-1/interleukin 10 (IL-10)-independent pathway [61]. As an inducible isoform of heme oxygenases, HO-1 catalyzes the oxidative conversion of heme to carbon monoxide (CO), ferrous ions, and biliverdin (BV), which is reduced to bilirubin-IXα (BR) by biliverdin reductase (BVR) [86]. Since heme can catalyze membrane lipid peroxidation and oxidize low-density lipoprotein, both BV and BR act as powerful antioxidants, and the removal of heme and production of BV and BR may responsible for the anti-oxidative effects of HO-1 [86]. The HO-1/CO system has been considered as a modulator of inflammation, apoptosis, and autophagy [87]; however, the involvement of HO-1-derived CO in the immunomodulatory effects of H_2_ needs to be further investigated.

Excessive ROS generation under abnormal metabolic conditions not only causes damage to cellular macromolecules, but also activates redox sensitive transcription factors, such as nuclear factor-kappaB (NF-kB), resulting in the release of pro-inflammatory mediators. Deregulated NF-kB activation is a hallmark of chronic inflammatory diseases [88]. Previous studies have provided evidence that H_2_ could alleviate inflammation via inhibition of NF-kB activation in metabolic diseases [16,32,65]. Since the induction of HO-1 expression has also been related to the inhibition of NF-kB activation [89], the activation of the Nrf2/HO-1 pathway induced by H_2_ may contributed to its anti-inflammatory effects. Thus, it is reasonable that the anti-inflammatory activities of H_2_ may be attributed to its antioxidant effects.

### 3.3. Anti-Apoptotic Effects of H_2_

Apoptosis or programmed cell death plays an important role in the maintenance of tissue homeostasis. Excess apoptosis has been implicated in the initiation and development of metabolic diseases, such as DM and its complications [90]. Extensive studies have demonstrated the anti-apoptotic effects of H_2_ in metabolic diseases, as evidenced by the upregulation of anti-apoptotic Bcl-2, downregulation of pro-apoptotic Bax and cleaved caspase-3, and inhibition of caspase activity, as well as reduced TUNEL positive cells [20,25,28,31,32,59,67]. Oxidative stress-induced DNA breakage can rapidly activate poly (ADP-ribose) polymerase-1 (PARP-1), a DNA nick sensing enzyme, leading to the induction of either caspase-independent apoptosis via translocation of apoptosis-inducing factor (AIF) from mitochondria to the nucleus or caspase-dependent apoptosis by activation of caspase-3. H_2_ has been reported to inhibit the activation of PARP-1, leading to a reduction in apoptosis of Schwann cells induced by high glucose-mediated oxidative stress via the caspase-independent and caspase-dependent pathways [25]. The inhibition of apoptosis by H_2_ could also be mediated by elevating HO-1-induced expression of an NAD-dependent protein deacetylase, Sirtuin1 (SIRT1) [59]. This elevated SIRT1 expression could further reduce the expression of Bax, acetylation of p53, and cleaved caspase-3, and subsequently suppress the hepatocyte apoptosis. In addition, Wu et al. reported that H_2_ could suppress the activation of the Jun NH2-terminal kinase (JNK) and p38 mitogen activated protein kinase (MAPK) signaling pathways, leading to a reduction of the Bax/Bcl-2 ratio and caspase-3 activity, and finally inhibiting apoptosis [32]. In fact, p53 is considered as a major JNK/p38 MAPK substrate in promoting apoptosis [91], thus the JNK/p38 MAPK/p53 pathway could also be involved in the suppression of apoptosis by H_2_. In addition, H_2_ could also inhibit apoptosis by suppressing the activation of the ER stress pathway [32,67]. In conclusion, H_2_ could inhibit the apoptosis induced by oxidative stress or other stimulus in metabolic diseases via multiple signaling pathways, including but not limited to the PARP-1-mediated caspase-dependent and caspase-independent pathways, and HO-1/SIRT1/p53, JNK/p38 MAPK, and ER stress pathways.

### 3.4. Suppression of ER Stress

Endoplasmic reticulum stress (ER stress) is a transient state of functional imbalance characterized by the accumulation of unfolded and misfolded proteins in the ER lumen, which is usually resolved by the activation of multiple adaptive mechanisms, such as the unfolded protein response (UPR), ER-associated protein degradation (ERAD), or autophagy, and that eventually induces two conflicting responses: survival or apoptosis [92]. As the “canonical ER stress response”, the UPR pathway is regulated by ER chaperone protein glucose-regulated protein 78 (GRP78), also known as binding immunoglobulin protein (BiP), and three traditional UPR sensors, including proteins R (PKR)-like endoplasmic reticulum kinase (PERK), inositol-requiring enzyme1α (IREα), and activating transcription factor 6 (ATF6) [93]. ER stress has been associated with a number of metabolic disorders, including T2DM, FL, AS, and obesity [92]. Previous studies have shown that H_2_ could alleviate ER stress, as evidenced by a reduction in the expression of C/EBP homologous protein (CHOP), X-box protein1 (XBP1), ATF6, phosphorylated PERK (P-PERK), and GRP78/BiP [DM-17, AS-4]. These results indicated that H_2_ may exert anti-ER stress effects via regulation of the UPR pathway. Whether the regulation of “non-canonical ER stress responses”, including integrated stress response (ISR), ER translocation and ERK reactivation, endoplasmic-reticulum-associated protein degradation (ERAD), and ER-phagy [93], is also involved in the suppression of ER stress by H_2_ requires further investigation. Furthermore, Nrf2 could also be activated by PERK-mediated phosphorylation, which coordinates the convergence of ER stress with oxidative stress signaling [94], and thus Nrf2 may serve as an important cross-talk mediator of the anti-oxidative pathways with the anti-ER stress pathways induced by H_2_.

### 3.5. Activation of Autophagy

Autophagy is an evolutionarily conserved intracellular lysosome-dependent catabolic process, involving turnover of long-lived proteins and damaged organelles, and thus contributing to maintenance of cellular homeostasis, clearance of damaged or excess cellular components, and adaptation to environmental challenges. Nutrient-sensing pathways, including the mechanistic target of rapamycin complex 1 (mTORC1), AMP-activated kinase (AMPK), and SIRT1, are involved in the regulation of autophagic flux, depending on nutrient status [95]. Insufficient levels of autophagy cause the accumulation of damaged organelles or lipid droplets, which is crucial for the pathogenesis of metabolic diseases [95]. H_2_ has been shown to activate autophagy in macrophages, to prevent AS development, as evidenced by an increased expression of LC3-II and autophagy-related (Atg) proteins Atg3 and reduced expression of p62 [68,69]. Yang et al. also provided evidence that SIRT1 is essential for the upregulation of autophagic flux by H_2_ in ox-LDL-treated macrophages [68]. Although defective autophagy has been widely implicated in the pathogenesis of metabolic diseases, upregulated autophagy in subcutaneous and visceral adipose tissue has also been observed in patients with obesity or T2DM [95]; whether H_2_ could inhibit autophagy responses in these conditions needs to be further investigated. In conclusion, the activation of autophagy by H_2_ may contribute to its beneficial effects in metabolic diseases; however, the detailed mechanism underlying the regulation of autophagic flux by H_2_ requires further investigation.

### 3.6. Improvement of Mitochondrial Function

Mitochondria, the cell powerhouses, are involved in essential cellular functions, including ATP production, intracellular Ca^2+^ regulation, production and modulation of ROS, and apoptosis [96]. Mitochondrial dysfunction leads to reduced ATP generation and mitochondrial biogenesis, and increased ROS and cytosolic Ca^2+^ imbalances, which have been linked to the pathogenesis of several metabolic disorders, including MS, DM, and obesity [96]. H_2_ has been widely demonstrated to inhibit ROS generation, directly scavenge ROS or indirectly removing ROS via enhancing antioxidant enzyme activity, which may be responsible for improvements of mitochondrial function in metabolic diseases. It has been shown that the Mito-K-ATP channels play a homeostatic role in blood glucose regulation in organisms, and their dysfunction has been associated with DM and its complications [97]. The activation of Mito-K-ATP channels could effectively prevent mitochondrial ROS production [98]. Jiao et al. demonstrated that 5-hydroxydecanoate, a selective Mito-K-ATP channel blocker, could eliminate the neuroprotective effects of H_2_ in diabetic peripheral neuropathy, indicating the activation of Mito-K-ATP channels by H_2_ [26]. In addition, Ma’s group provided evidence that H_2_ could promote fatty acid oxidation, probably by inducing their transport to mitochondria and subsequent catabolism to ketone bodies, as evidenced by an increased level of acylcarnitines, the mitochondrial fatty acid transporters, and acetoacetate, a marker of ketogenesis in the liver, which may be mediated by the NADP/NADPH redox pathway [99]. In conclusion, improvement of mitochondrial function by H_2_, including but not limited to the activation of Mito-K-ATP channels and promotion of mitochondrial fatty acids oxidation, may be involved in the beneficial effects of H_2_ in metabolic diseases.

### 3.7. Regulation of Gut Microbiota

The gut microbiota is a dense and diverse microbial ecosystem dominated by bacteria, which mainly comprise four main phyla: *Actinobacteria*, *Bacteroidetes*, *Firmicutes,* and *Proteobacteria* in mammalians [100]. Gut microbiota-derived metabolites can interact with the host and serve as central regulators in metabolic disorders, including T2DM, NAFLD, and obesity [101]. Qiu et al. reported that HRS could alleviate HFD-induced dysbacteriosis by inhibiting the glyoxylic acid cycle of the intestinal flora [72]. He’s group found that sustained and ultrahigh H_2_ intervention in MAFLD mice drastically changed the composition of gut microbiota, reduced the ratio of *Firmicutes*/*Bacteroidetes* (F/B), and increased the abundance of *Akkermansia muciniphila* (*Akk.m*) [62]. The *Akk.m* has been considered a gut probiotic that can improve metabolic diseases, and its genome contains genes encoding hydrogenases, enabling it to catalytically decompose and utilize H_2_. It is likely that the utilization of H_2_ contributes to the outgrowth of *Akk.m*, finally leading to the improvement of MAFLD. It is worth noting that H_2_ could improve glucose metabolism but not lipid and energy metabolism via reprogrammed gut microbiota, because antibiotic treatment abolished the blood glucose-lowering effect but did not affect lipid parameters. In conclusion, bacterial hydrogenases enable the utilization of exogenous H_2_, leading to an altered gut microbiome profile, which may eventually regulate the host metabolism, especially the glucose metabolism, through gut microbiota-derived metabolites.

### 3.8. Other Possible Mechanisms and Potential Molecular Targets of H_2_

In addition to the above mechanisms underlying the effects of H_2_ on metabolic diseases, previous studies have also explored the possibility of other potential mechanisms. Qin’s group demonstrated that H_2_ could improve the protein composition and functionality of HDL by modulating the oxidized phospholipids (OxPLs)-related enzymes binding to HDL [74]. Zou et al. Provided evidence that H_2_ could reduce pyroptosis, a type of programmed cell death that occurs in the presence of inflammation, via inhibition of the AMPK/mTOR/NLRP3 signaling pathway in diabetic cardiomyopathy [33]. Li et al. reported that H_2_ could protect Schwann cells from high glucose-induced parthanatos, which is a type of PARP-1-dependent programmed cell death [27]. Although many possible mechanisms of H_2_ have been proposed, the target molecule of H_2_ is still unclear. Ohta’s group first considered hydroxyl radicals and peroxynitrite as the potential targets of H_2_; however, this view has been debated, due to the considerably low reaction rate of H_2_ with hydroxyl radicals. Later, several research groups proposed that some protein molecules may be potential targets of H_2_. Ma’s group first provided evidence that H_2_ could directly interact with acetylcholinesterase (AChE) and enhance its activity [102]. They later found that horseradish peroxidase (HRP) might also be the target molecule of H_2_. H_2_ might induce alterations in the microenvironment of histidine, tryptophan, and aspartate in HRP, and reconstruct the hydrogen bond network with the “H_2_O-H_2_O” bridge, thus increasing the efficiency of energy transfer to the heme center, and finally enhancing the enzyme activity [103]. Another potential molecular target of H_2_ was identified by He’s group. They reported that Fe-porphyrin may serve as a redox-related biosensor of H_2_, making it the upstream signaling molecular of CO through Fe-porphyrin-catalytic reduction of CO_2_ into CO in the hypoxic microenvironment [104]. Fe-porphyrin is a common heme cofactor in the active sites of many proteins, including hemoglobin (Hb) and cytochromes (Cyt), as well as HRP. They also provided evidence of the catalytic hydrogenation activity of Fe-porphyrin on the levels of proteins and cells, such as the catalytic activity of Hb for the hydrogenation of hydroxyl radicals and the reduction of CO_2_ by H_2_ into CO through Fe-porphyrin–catalytic hydrogenation. Heme proteins can also interact with other gaseous molecules, such as CO, NO, O_2_, and H_2_S, which play vital roles in many biological processes [105,106]. Thus, it is likely that H_2_ competes with them to interact with heme proteins, which ultimately affects the physiological functions of these gaseous molecules. Ohta recently proposed that the oxidative potential of hydroxyl radicals may be alleviated by H_2_-targeting porphyrin, which contributes to the activation of Nrf2 as a hormesis-like effect [107]. In conclusion, H_2_ may interact with multiple target molecules, including but not limited to hydroxyl radicals, peroxynitrite, AChE, HRP, and Fe-porphyrin, which finally contribute to the extensive biological effects of H_2_.

## 4. Conclusions and Perspectives

To date, both pre-clinical and clinical studies have provided evidence that H_2_ interventions might exert beneficial effects on metabolic diseases, with no reported adverse effects. However, the efficacy of intervention varied among the different studies, with some studies even showing no therapeutic effects. There are serval factors that may contribute to these differences: (1) H_2_ concentration—the dose-dependent effect of H_2_ has been observed in certain studies, which showed that higher doses of H_2_ exhibited better therapeutic effects. Different H_2_ delivery methods may cause different H_2_ concentrations in organs related to the initiation and development of metabolic diseases, and ultimately lead to different treatment effects. (2) Treatment duration—the response time to H_2_ may vary in different diseases, such as the relative faster regulation of hyperlipemia than hyperglycemia by H_2_. Prolonged H_2_ treatment may have a more significant impact on metabolic diseases. (3) The time point of H_2_ intervention—the effectiveness of H_2_ therapy may differ in different stage of metabolic diseases. (4) The severity of metabolic disorders— pre-clinical studies have shown that higher initial FBG levels prior to intervention seem to have a more apparent FBG-lowering effect. However, the current research only provides preliminary experimental evidence of the influence of the above factors on the effect of H_2_ intervention, and more in-depth and detailed research should be carried out in the future. Furthermore, an increasing number of clinical trials have been designed to evaluate the therapeutic effect of H_2_ on metabolic diseases; however, high-quality clinical studies have been relatively limited, and large-scale, randomized, controlled, multicentered, double-blind, and long-term interventions should be considered in future clinical trial designs.

The mechanism of action of H_2_ has long been a focal point and challenge for research. Although many hypotheses of H_2_ action have been proposed, most can be attributed to the antioxidative effect of H_2_. Free radicals have been considered as a double-edged sword in various disease treatments [108]; even hydroxyl radicals, the most cytotoxic of ROS, do not always have negative effects on cells. For example, the superoxide anion and its derivative hydroxyl radicals are essential for human phagocytes, to kill invading bacteria [109]. Thus, the selective hydroxyl radical-scavenging activity of H_2_ cannot explain its extensive beneficial effects. From the current research, H_2_ may have multiple target molecules, and the phenotypes induced by H_2_ intervention may be a comprehensive result of the interaction between molecular hydrogen and its target molecules. Future mechanistic studies should be focused on the identification of the potential targets of H_2_ and on exploring how H_2_ interacts with target molecules to induce cellular responses.

In conclusion, the evidence from the current studies suggests the potential application of H_2_ in the treatment of metabolic diseases. It is believed that, with further clinical and mechanism research, H_2_ will eventually be applied to clinical practice in the future, to benefit more patients.

## Figures and Tables

**Figure 1 pharmaceuticals-16-00541-f001:**
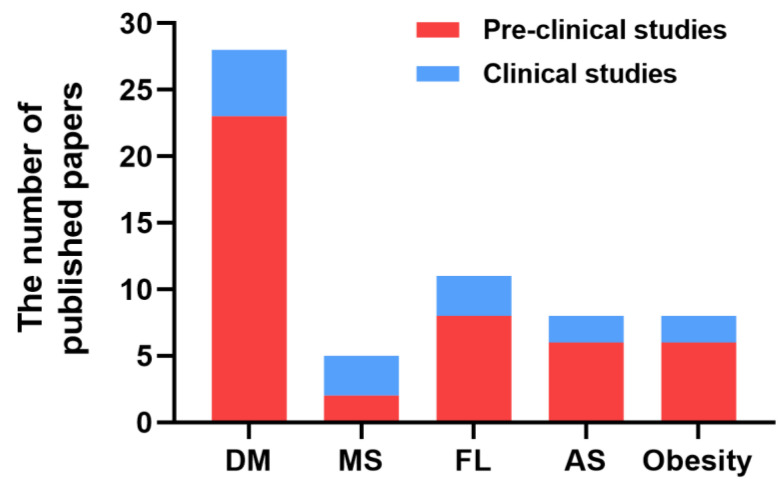
The number of papers published on the effects of H_2_ on metabolic diseases.

**Figure 2 pharmaceuticals-16-00541-f002:**
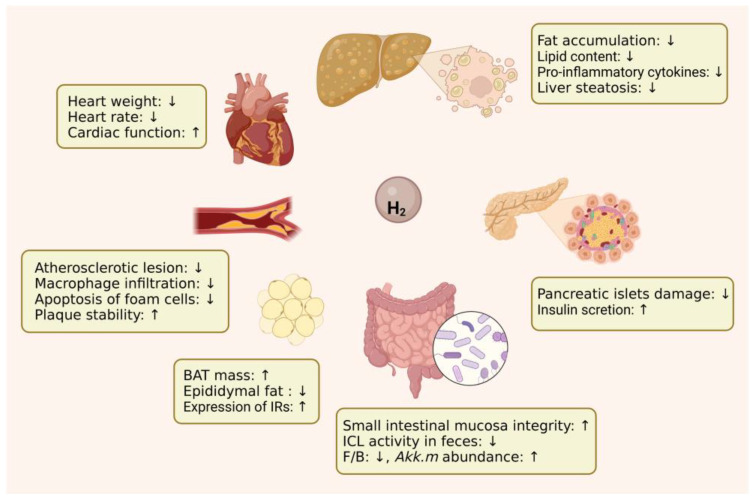
The main effects of H_2_ on various tissues in metabolic diseases. Abbreviations: BAT, brown adipose tissue; IRs: insulin receptors; ICL: isocitrate lyase; F/B, *Firmicutes*/*Bacteroidetes*; *Akk.m*, *Akkermansia muciniphila*.

**Figure 3 pharmaceuticals-16-00541-f003:**
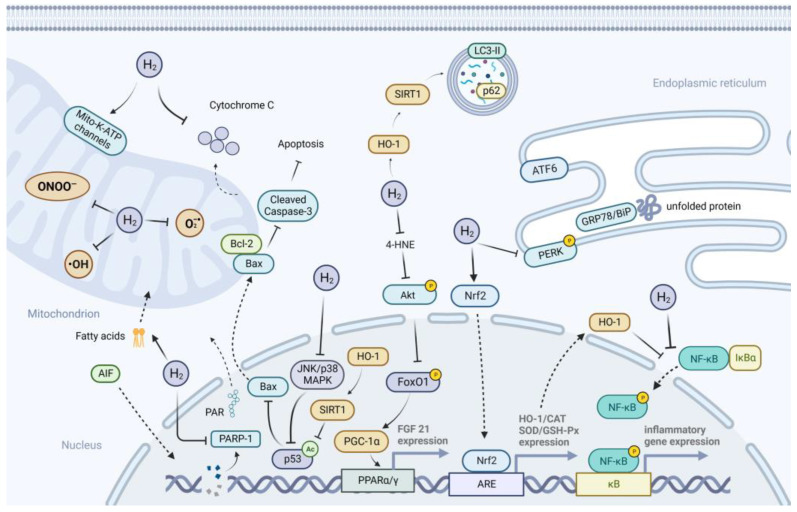
The possible mechanisms underlying the effects of H_2_ on metabolic diseases. The figure demonstrates the potential mechanisms underlying the effects of H_2_ on metabolic diseases, including scavenging or inhibiting generation of ROS; activating Mito-K-ATP channels; promoting the transport of fatty acids into mitochondria; as well as the pathways involved in the anti-oxidative, anti-inflammatory, and anti-apoptotic effects, and suppression of ER stress and activation of autophagy, including the Nrf2/ARE/HO-1, PARP-1/AIF, HO-1/SIRT1/p53, JNK/p38 MAPK/p53, Akt/FoxO1/PGC-1α/PPARα/γ, NF-kB-mediated, and PERK-mediated UPR pathways. Abbreviations: Mito-K-ATP, mitochondrial ATP-sensitive potassium; SIRT1, Sirtuin1; LC3-II, microtubule-associated protein light chain 3-II; HO-1, heme oxygenase-1; ATF6, activating transcription factor 6; GRP78/BiP, glucose-regulated protein 78/binding immunoglobulin protein; PERK, proteins R (PKR)-like endoplasmic reticulum kinase; 4-HNE, 4-hydoxy-2-nonenal; FoxO1, forkhead box O1; PGC-1α, peroxisome proliferator-activated receptor gamma coactivator-1 alpha; PPARα/γ, peroxisome proliferator-activated receptor alpha/gamma; Nrf2, nuclear factor erythroid 2-related factor 2; ARE, antioxidant responsive element; NF-kB, nuclear factor-kappaB; IkBα, inhibitory subunit of NF-kB alpha; JNK/p38 MAPK, c-Jun N-terminal kinase/p38 mitogen-activated protein kinase; Bcl-2, B-cell lymphoma 2; Bax, Bcl-2-associated X protein; PARP-1, poly (ADP-ribose) polymerase-1; PAR, poly(ADP-ribose); AIF, apoptosis-inducing factor; CAT, catalase; SOD, superoxide dismutase; GSH-Px, glutathione peroxidase; FGF21, fibroblast growth factor 21.

**Table 1 pharmaceuticals-16-00541-t001:** The effects of H_2_ intervention on DMs.

Disease Type	Model System	Routes of H_2_ Delivery	H_2_ Concentration	Duration	Initial FBG Levels (mmol/L)	Reduction (%)	Main Effects	Reference
T2DM	HFD (4 weeks) and low-dose STZ (25 mg/kg)-induced DM rat	drinking HRW	>0.6 mM	3 weeks	~22	~14	FBG, TG, TC, LDL-c, HDL-c, IL-1β, hepatic fat, renal and spleen tissue damage: ↓ GHb: NSC	[12]
T2DM	HFD (4 weeks) and STZ (100 mg/kg)-induced DM mice	subcutaneous injection of H_2_ gas	~0.8 mM	4 weeks	~24	~40	FBG, insulin, TG, LDL-c: ↓ HDL-c: ↑ glucose tolerance, insulin sensitivity: ↑ diabetic renal injury: ↓	[13]
T2DM	HGHFD-induced (8 weeks) insulin resistance rat model, high glucose and HFD (4 weeks) and low-dose STZ (20 mg/kg)-induced DM rat	oral gavage of HRS	NA	8 weeks	NA	IR: 12.50%; DM: 19.63%	IR: FBG, insulin, TG, TC, LDL-c: ↓, insulin sensitivity: ↑ DM: FBG, TG, TC, LDL-c: ↓	[14]
T2DM	HFD (30 days) and low-dose STZ (35 mg/kg)-induced DM rat	drinking HRW	0.5 mM	2 weeks	NA	NA	insulin: ↑ IRs expression in the adipose and skeletal muscle tissues: ↑	[15]
T2DM	HFD (4 weeks) and low-dose STZ (30 mg/kg)-induced DM rat	intravenous injection of HRS	>0.6 mM	80 days	~28	~50	FBG, TG, TC, LDL-c: ↓ HDL-c: ↑ insulin resistance, pancreatic islets and glomeruli damage: ↓ insulin: NSC	[16]
T2DM	Lepr^*db*/*db*^ mice	drinking HRW	0.8 mM	3 months	NA	20.75	FBG, insulin, TG, hepatic fat: ↓	[17]
T2DM	HFD-induced mice, Lepr^*db*/*db*^ mice	drinking HRW	0.8 mM	HFD-induced mice: 25 weeks; db/db mice: 18 weeks	HFD-induced mice): 9.61; db/db mice: 16.56	NA	FBG, GA, lipid parameters: NSC;	[18]
T1DM	STZ (50 mg/kg/day for 5 days)-induced T1DM mice	drinking HRW or intraperitoneal injection of HRS	0.8 mM	HRW: 18 weeks; HRS: 4 weeks	NA	HRS: 19.31; HRW: NSC	HRS: FBG, GHb, TG: ↓ HRW: FBG: NSC; GHb, TG: ↓	[18]
T2DM	Lepr^*db*/*db*^ mice	drinking ERW	0.15–0.3 mM	4 weeks	~9	~41	FBG: ↓ insulin: ↑; glucose tolerance: NSC	[19]
T1DM	STZ (60 mg/kg/day for 5 days)-induced T1DM mice	drinking ERW	0.15–0.3 mM	6 weeks	NA	38.98	FBG: ↓ insulin: NSC glucose tolerance: ↑	[19]
T1DM	Alloxan (100 mg/kg/injection, 3 injections)-induced T1DM mice	drinking ERW	~0.5 mM	9 weeks	5.39	56.35	FBG: ↓ insulin: ↑	[20]

Abbreviations: HFD, high-fat diet; STZ, streptozotocin; FBG, fast blood glucose; TG, total triglyceride; TC, total cholesterol; LDL-c, low-density lipoprotein cholesterol; HDL-c, high-density lipoprotein cholesterol; IL-1β, interleukin-1 beta; GHb, glycated hemoglobin; HGHFD, high glucose and high fat diet; IR, insulin resistance; IRs, insulin receptors; Lepr^*db*/*db*^, leptin receptor-deficient db/db; GA, glycated albumin; ERW, electrolyzed-reduced water; NSC, no significant change; NA, not available.

**Table 2 pharmaceuticals-16-00541-t002:** The effects of H_2_ intervention on metabolic diseases in clinical trials.

Disease/Objective	Subjects No.		Routes of H_2_ Delivery	H_2_ Concentration	Duration	Main Effects	Clinical Registration No.	Reference
	Male	Female						
T2DM/IGT	18	18	drinking HRW	~0.6 mM	8 weeks	sdLDL, emLDL, u-IsoP: ↓, oxLDL: ⇓ glucose tolerance in 4 of 6 IGT patients: ↑ TG, TC, LDL, HDL, FBG, HbA1C, insulin: NSC	NA	[39]
T2DM	25	20	drinking ERW	NA	12 weeks	lactate: ↓, FBG, insulin, HOMA-IR: NSC the rate of change in lactate was positively correlated with the rate of change in HOMA-IR, FBG and insulin in the HOMA-IR ≥1.73 group, in the FBG ≥6.1 mM group: insulin: ↓, FBG: ⇓	UMIN000019032	[40]
T2DM	15	15	drinking ERW	NA	12 days	TC, TG, LDL: ↓, HDL: ↑ FBG: ⇓	NA	[41]
MS	10	10	drinking HRW	0.55–0.65 mM	8 weeks	SOD: ↑, TBARS in urine: ↓ HDL: ↑ (week 4) LDL, TC/HDL: ↓ (week 4) TC, TG, FBG: NSC	NA	[42]
MS	12	8	drinking HRW	0.2–0.25 mM	10 weeks	TC, LDL, apoB100, apoE: ↓ TG, HDL, FBG: NSC MDA: ↓, SOD: ↑ TNF-α, IL-6: NSC	NA	[43]
MS	30	30	drinking high-concentration HRW (>5.5 mM)	>5.5 mmol H2 per day	24 weeks	BMI, WHC, TG, TC, LDL, HDL, FBG, HbA1c: ↓ TNF-α, IL-6, CRP: ↓ MDA: ↓, TBARS: NSC Vitamin E, Vitamin C: ↑	NA	[44]
NAFLD	5	7	drinking HRW	3 mM	4 weeks	liver fat, AST: ↓ BW, BMI, body composition, lipid profiles, FBG: NSC	NCT03625362	[45]
NAFLD	13	17	drinking HRW	>2 mM	8 weeks	BW, BMI, SBP, AST, ALT, LDH, NF-κB, HSP70, MMP-9: ⇓ CRP, ALB, ALP, TG, TC, HDL, LDL, TG/HDL ratio, 8-OHdG, MDA: ⇑	NCT05325398	[46]
NAFLD	24	19	Hydrogen/oxygen inhalation	66%	13 weeks	TC, TG, HDL: NSC LDL, AST, ALT, MDA, TNF-α, IL-6: ↓ SOD: ↑ liver fat in moderate–severe cases: ↓	ChiCTR-IIR-16009114	[47]
Vascular endothelial function	18	16	drinking high-concentration HRW (3.5 mM)	>3.5 mM	1 time	the ratio of the changes in FMD of the BA: ↑	NA	[48]
Vascular endothelial function	24	44	drinking high-concentration HRW (3.5 mM)	>3.5 mM	2 weeks	Ln_RHI value in the low Ln_RHI group: ↑ Ln_RHI value in the high Ln_RHI group: NSC	NA	[49]
Obesity	0	10	oral intake of H2 generating minerals	~6 ppm of H2 per day	4 weeks	body fatness, arm fat index, TG, insulin: ↓ ghrelin: ⇑, lactate: ↓ BW, FBG, other lipid parameters: NSC	NCT02832219	[50]
Obesity	2	2	HRW bath (41 °C, 0.15 mM)	~0.15 mM	4–24 weeks	visceral fat, LDL in two women: ↓	NA	[51]

Abbreviations: IGT, impaired glucose tolerance; sdLDL, small dense LDL; emLDL, electronegative charge of modified LDL; u-IsoP, urinary 8-isoprostanes; oxLDL, oxidized LDL; HbA1C, hemoglobin A1c; HOMA-IR, homeostasis model assessment of insulin resistance; TBARS, thiobarbituric acid reactive substances; apoB100, apolipoprotein B100; apoE, apolipoprotein E; TNF-α, tumor necrosis factor alpha; IL-6, interleukin-6; BMI, body mass index; WHC, waist–hip circumference; CRP, C-reactive protein; MDA, malondialdehyde; AST, aspartate aminotransferase; BW, body weight; SBP, systolic blood pressure; ALT, alanine transferase; LDH, lactate dehydrogenase; NF-κB, nuclear factor kappa B; HSP70, heat shock protein 70; MMP-9, matrix metalloproteinase-9; ALB, albumin; ALP, alkaline phosphatase; 8-OHdG, 8-hydroxydeoxyguanosine; FMD, flow-mediated dilation; BA, brachial artery; Ln_RHI, the natural logarithmic transformed value of the RH-PAT (reactive hyperemia-peripheral arterial tonometry) index; ⇓, decrease without significance; ⇑, increase without significance; NSC, no significant change; NA, not available.

**Table 3 pharmaceuticals-16-00541-t003:** The effects of H_2_ intervention on MS, FL, AS, and obesity.

Disease Type	Model System	Routes of H_2_ Delivery	H_2_ Concentration	Duration	Main Effects	Reference
MS	SHR-cp rats	drinking HRW	0.15–0.2 mM	16 weeks	plasma BUN, creatinine:↓, BAP: ↑ plasma TG, TC, FBG, 8-OHdG: NSC, glomerulosclerosis score: ↓ 24 h water intake and urine flow: ↑ urinary albumin to creatinine ratio: ↓	[53]
MS	SHR-cp rats	drinking HRW	0.15–0.2 mM	16 weeks	renal glyoxal, methylglyoxal, and 3-deoxyglucosone levels: ↓ ROS production in kidney: ↓	[55]
ALD	chronic-binge ethanol-fed mice model	drinking HRW (0.25 mM)	0.25–0.3 mM	13 weeks	EtOH-induced anorexia and liver enlargement: ↓ serum TG, TC, ALT, TNF-α, IL-6, oxidative stress: ↓, IL-10, IL-22: ↑ hepatic TG, TC, TNF-α, IL-6: ↓	[56]
NAFLD	STZ (25 mg/kg, single dose) + HFHSD (8 weeks)-induced rats model	intraperitoneal injection of HRS	>0.6 mM	8 weeks	serum ALT, TBIL, TC, TG, FBG, insulin: ↓ insulin sensitivity, glucose tolerance: ↑ hepatocyte apoptosis: ↓ Inflammation, oxidative stress: ↓	[57]
NAFLD	HFD-induced mice model	drinking HRW	0.8 mM	8 weeks	serum ALT, AST, TC, TG: ↓ hepatic expression of MEG3, Nrf2: ↑, miR-136: ↓	[58]
NAFLD	MCDHF diet-induced mice model	HRS (3.5 mM) spread in the liver after clipping hepatic portal vein with a microvascular clamp for ischemia	3.5 mM	1 time	serum ALT, AST: ↓ ischemia–reperfusion injury in fatty liver: ↓ hepatic expression of TNF-α, IL-6, TLR-4, Nlrp3: ↓	[59]
NAFLD	HFFD-induced rats model	hydrogen inhalation	4% or 67%	10 weeks	BW, BMI, abdominal fat index, liver index: ↓ glucose tolerance: ↓ serum TG, ALT, AST, LDH: ↓, TC: NSC hepatic fat: ↓	[60]
NASH and fibrosis	CDAA diet (20 weeks)-induced mice model	drinking HRW	3.5 mM	4, 8, or 20 weeks	serum AST, ALT, TBARS: ↓ hepatic fibrosis, macrophage recruitment, HSC activation, expression of inflammatory cytokines: ↓	[61]
NAFLD	HFD (16 weeks)-induced mice model and db/db mice (6 weeks)	oral intake of hydrogen nanocapsule-mixed HFD	>10^4^ times higher than that of HRW	16 or 6 weeks	BW, WAT mass, liver weight, liver steatosis: ↓ plasma TC, FBG, insulin: ↓, glucose tolerance: ↑ food intake, plasma TG: NSC	[62]
NASH and HCC	MCD diet (8 weeks)-induced mice model and HFD (8 weeks) induced-STAM^®^ mice model	drinking HRW	0.175–0.225 mM	8 weeks	plasma ALT, hepatic TC, plasma and hepatic oxidative stress, hepatocyte apoptosis, NAS: ↓ hepatic expression of free fatty acid uptake-related enzymes, inflammatory cytokines, PPARα: ↓ hepatocarcinogenesis: ↓	[63]
AS	apoE knockout mice model	drinking HRW	>0.6 mM	8 weeks	atherosclerotic lesion: ↓ oxidative stress level of aorta: ↓	[64]
AS	apoE knockout mice model	intraperitoneal injection of HRS	>0.6 mM	8 weeks	LOX-1 expression and NF-κB activation in thoracic aorta: ↓	[65]
AS	apoE knockout mice model	intraperitoneal injection of HRS	>0.6 mM	8 weeks	plasma TC, non-HDL-C, MDA, SAA: ↓, PON-1: ↑ plasma and hepatic apoB: ↓ aortic inflammation: ↓ RCT-related genes expression: ↑ oxidation of non-HDL: ↓, HDL quality: ↑	[66]
AS	Lepr^*db*/*db*^ mice fed an atherogenic diet	intraperitoneal injection of HRS	>0.6 mM	28 weeks	atherosclerotic plaque stability: ↑ the formation and apoptosis of macrophage-derived foam cells: ↓ serum ox-LDL, ROS in aorta, peritoneal macrophages: ↓	[67]
AS	mouse macrophage-like cell line	cultured in HRM	0.6 mM	24 h	ox-LDL-induced inflammation: ↓ autophagic flux: ↑	[68]
AS	apoE knockout mice model	intravenously injected with tetrapod needle-like PdH nanozyme loaded by macrophages	NA	8 weeks	plaque area and the necrotic cores in aortic sinus sections: ↓ macrophage infiltration and MMP-9 expression in atherosclerotic plaques: ↓ stability of atherosclerotic plaques: ↑	[69]
obesity	Lepr^*db*/*db*^ mice, HFD-induced obesity mice	drinking HRW	0.8 mM	12 weeks	hepatic oxidative stress and lipid accumulation: ↓ BW, serum FBG, TG, insulin: ↓ TC, LDL, HDL: NSC	[17]
obesity	HFD-induced obesity mice	drinking ERW	0.05 mM	12 weeks	BW, epididymal fat, liver fat: ↓ blood neutrophils and lymphocytes: ↑ serum adiponectin: ↑	[70]
obesity	HFD-induced obesity mice	gavage of MgH_2_ suspension	NA	74 weeks	plasma TG: ↓ the average of lifespan: ↑	[71]
obesity	HFD-induced mice	gavage of HRS	>0.6 mM	2, 4, 6 weeks	BW, blood TC, TG, LDL: ↓, HDL: ↑ intestinal integrity: ↑ abundance of Bacteroides, Bifidobacteria, Lactobacillus in feces: ↑, Enterobacter cloacae: ↓, ICL activity: ↓	[72]
obesity	HFD-induced mice	drinking HRW	~0.45 mM	2 weeks	heart weight: ↓, BAT mass: ↑ cardiac function, vascular bioactivity: ↑ FBG, BW, WAT mass, HR: NSC	[73]
obesity	HFD-induced rats	hydrogen inhalation	4%	10 weeks	hepatic FFA, MDA, 4-HNE, PGPC, PONPC, PAzPC: ↓, TC, TG: ⇓ plasma PONPC: ↓, TC, TG, FFA, MDA, 4-HNE: NSC plasma HDL antioxidant activity: ↑, Lp-PLA2 activity: ↓, PON-1 activity, LCAT expression: ↑	[74]

Abbreviations: SHR-cp, SHR.Cg-Leprcp/NDmcr; BUN, blood urea nitrogen; BAP, biological antioxidant potential; EtOH, ethyl alcohol; IL-10, interleukin-10; IL-22, interleukin-22; HFHSD, high fat and high sugar diet; TBIL, total bilirubin; MEG3, maternally expressed gene 3; Nrf2, nuclear factor erythroid 2-related factor 2; miR-136, microRNA-136; MCDHF, methionine and choline-deficient plus high fat; TLR-4, toll-like receptor-4; Nlrp3, nucleotide-binding oligomerization domain (NOD)-like receptor 3; HFFD, high fat and fructose diet; NASH, nonalcoholic steatohepatitis; CDAA, choline-deficient, L-amino acid-defined; HSC, hepatic stellate cell; WAT, white adipose tissue; MCD, methionine and choline-deficient; NAS, NAFLD activity score; PPARα, peroxisome proliferator-activated receptor alpha; LOX-1, lectin-like oxidized LDL receptor-1; SAA, serum amyloid A; PON-1, paraoxonase-1; RCT, reverse cholesterol transport; HRM, hydrogen-rich medium; ICL, isocitrate lyase; BAT, brown adipose tissue; HR, heart rate; FFA, free fatty acids; 4-HNE, 4-hydoxy-2-nonenal; PGPC, 1-palmitoyl-2-glutaroyl-sn-glycero-3-phosphatidylcholine; PONPC, 1-palmitoyl-2-(9-oxo-nonanoyl)-sn-glycero-3-phosphatidylcholine; PAzPC, 1-palmitoyl-2-azelaoyl-sn-glycero-3-phosphatidylcholine; Lp-PLA2, lipoprotein-associated phospholipase A2; LCAT, lecithin:cholesterol acyltransferase; ⇓, decrease without significance; NSC, no significant change.

## Data Availability

Data presented in this study are available on request from the corresponding author.

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
