# Peer review of "Therapeutic Potential of Molecular Hydrogen in Metabolic Diseases from Bench to Bedside"

_pharmaceuticals, 2023, doi:10.3390/ph16040541_

Round 1
Reviewer 1 Report
The presented article by authors Xie et al. summarizes the current scientific knowledge about the molecular hydrogen effect on various metabolic diseases, where the authors have described experiments on cells, animals, and humans in a very appropriate and logical way. Molecular hydrogen is a new potential therapeutic substance with pluripotent effect on many mostly oxidative stress-induced diseases and thus it is actual and attractive topic.
This is a very well-written, comprehensive, up-to-date, and interesting review with minimal grammatical or stylistic issues. Manuscript is clear and easy to read. The authors very aptly summarize the results of many studies dealing with metabolic diseases and molecular hydrogen as a treatment. A great benefit is the presentation of clinical studies/trials from metabolic diseases research.
I have only small comments and recommendations:
- The manuscript contains a lot of abbreviations but not all of them are explained, e.g., on page 3 (MDA, GSH, SOD, CAT), table 1 and 2 (FBS), page 13 (FINS), etc. The abbreviations should be checked in the whole manuscript and further explained where necessary.
- The manuscript also contains a few typing errors, like MALFD (p. 20), drinking ARW (table 3), EHW (table 2). Please, correct these in the whole manuscript.
- In the tables, the authors present a lot of studies with molecular hydrogen administration. According to my opinion, it would be very beneficial if the authors also indicated the concentrations of hydrogen used in the works cited. They state the concentrations of H2 in some, but not in all of them.
In conclusion, according to previously mentioned issues and my comments, I recommend this article to be accepted after minor revision for publication in this journal.
Author Response
Reviewer #1: The presented article by authors Xie et al. summarizes the current scientific knowledge about the molecular hydrogen effect on various metabolic diseases, where the authors have described experiments on cells, animals, and humans in a very appropriate and logical way. Molecular hydrogen is a new potential therapeutic substance with pluripotent effect on many mostly oxidative stress-induced diseases and thus it is actual and attractive topic.
This is a very well-written, comprehensive, up-to-date, and interesting review with minimal grammatical or stylistic issues. Manuscript is clear and easy to read. The authors very aptly summarize the results of many studies dealing with metabolic diseases and molecular hydrogen as a treatment. A great benefit is the presentation of clinical studies/trials from metabolic diseases research.
I have only small comments and recommendations:
- The manuscript contains a lot of abbreviations but not all of them are explained, e.g., on page 3 (MDA, GSH, SOD, CAT), table 1 and 2 (FBS), page 13 (FINS), etc. The abbreviations should be checked in the whole manuscript and further explained where necessary.
Response: We have added the missing explanation of these abbreviations, e.g. MDA, GSH, SOD, CAT, FINS, IL-1β, apoB100, apoE, TNF-α, IL-6, TLR-4, Nlrp3, et al. FBS (fasting blood sugar) and FBG (fasting blood glucose) are the same, so we replaced FBS with FBG.
- The manuscript also contains a few typing errors, like MALFD (p. 20), drinking ARW (table 3), EHW (table 2). Please, correct these in the whole manuscript.
Response: “MALFD” has been replaced with “MAFLD”. ERW (electrolyzed-reduced water) is also called ARW (alkaline reduced water) or EHW (electrolyzed hydrogen-rich water), so we replaced ARW or EHW with ERW.
- In the tables, the authors present a lot of studies with molecular hydrogen administration. According to my opinion, it would be very beneficial if the authors also indicated the concentrations of hydrogen used in the works cited. They state the concentrations of H2 in some, but not in all of them.
Reponse: we have added the concentrations of hydrogen used in the works cited in the revised manuscript.
In conclusion, according to previously mentioned issues and my comments, I recommend this article to be accepted after minor revision for publication in this journal.
Reviewer 2 Report
Comments and suggestions
Thanks for your review manuscript entitled" Therapeutic potential of molecular hydrogen in metabolic diseases from bench to bedside'' I read this manuscript with great interest and felt it needs some modifications for improve of your manuscript for considering in this reputed journal. I am sharing my comments and suggestions details below:
1. In abstract section authors demonstrate that Oxidative stress has been implicated in the pathophysiology of metabolic diseases, including diabetes mellitus (DM), metabolic syndrome (MS), fatty liver (FL), atherosclerosis (AS), 12 and obesity. Molecular hydrogen (H2) has long been considered as a physiologically inert gas. Only oxidative stress is the the major pathophysiology role of metabolic diseases? Please check and confirm.
2. With further clinical and mechanism research, it is believed that H2 will be eventually applied to clinical practice in the future to benefit more patients. Please indicate which type of clinical trial and what kind of disease can be applied H2 therapy in future.
3. Introduction should be more expanded related metabolic diseases, including diabetes mellitus (DM), metabolic syndrome (MS), fatty liver (FL), atherosclerosis (AS), and obesity relation with H2. Present introduction is not sufficient to prove your evidence according to your title.
4. Possible Mechanisms of H2 Action on Metabolic Diseases. In figure 3 authors need more explanation relation H2 and metabolic disease with recent references. Also need to increase figure resolution. Present figure is obscure. It should be improved during revision stage.
5. Other possible mechanisms and the potential molecular targets of H2. Please expand this section to improve your manuscript with relevant study references. Because this one of the possible hypothesis of your manuscript.
6. Many references used very old throughout of the manuscript . Please revise more recent references relevant with your study.
7. Please check all syntax, typo and grammatical errors throughout of the manuscript.
Author Response
Reviewer #2: Thanks for your review manuscript entitled" Therapeutic potential of molecular hydrogen in metabolic diseases from bench to bedside'' I read this manuscript with great interest and felt it needs some modifications for improve of your manuscript for considering in this reputed journal. I am sharing my comments and suggestions details below:
- In abstract section authors demonstrate that Oxidative stress has been implicated in the pathophysiology of metabolic diseases, including diabetes mellitus (DM), metabolic syndrome (MS), fatty liver (FL), atherosclerosis (AS), 12 and obesity. Molecular hydrogen (H2) has long been considered as a physiologically inert gas. Only oxidative stress is the major pathophysiology role of metabolic diseases? Please check and confirm.
Response: we have revised as “Oxidative stress and chronic inflammation have been implicated in the pathophysiology of metabolic diseases”
- With further clinical and mechanism research, it is believed that H2 will be eventually applied to clinical practice in the future to benefit more patients. Please indicate which type of clinical trial and what kind of disease can be applied H2 therapy in future.
Response: we have revised as “With more high-quality clinical trials and in-depth mechanism researches, it is believed that H2 will be eventually applied to clinical practice in the future to benefit more patients with metabolic diseases.”
- Introduction should be more expanded related metabolic diseases, including diabetes mellitus (DM), metabolic syndrome (MS), fatty liver (FL), atherosclerosis (AS), and obesity relation with H2. Present introduction is not sufficient to prove your evidence according to your title.
Response: we have revised as “A considerable amount of literature has been published on the potential beneficial effects of H2 intervention on metabolic diseases so far (Figure 1), including improving glucose and lipid metabolism, alleviating pancreatic beta-cell damage, attenuating hepatic steatosis, promoting atherosclerotic plaque stabilization, etc.”
- Possible Mechanisms of H2 Action on Metabolic Diseases. In figure 3 authors need more explanation relation H2 and metabolic disease with recent references. Also need to increase figure resolution. Present figure is obscure. It should be improved during revision stage.
Response: we have added the explanation for figure 3 as “The figure demonstrates the potential mechanisms underlying the effects of H2 on metabolic diseases, including scavenging or inhibiting generation of ROS, activating Mito-K-ATP channels, promoting the transport of fatty acids into mitochondria, as well as the pathways involved in the anti-oxidative, anti-inflammatory, and anti-apoptotic effects, suppression of ER stress and activation of autophagy, including Nrf2/ARE/HO-1, PARP-1/AIF, HO-1/SIRT1/p53, JNK/p38 MAPK/p53, Akt/FoxO1/PGC-1α/PPARα/γ, NF-kB-mediated and PERK-mediated UPR pathways.” The resolution of the figures have also been impoved.
- Other possible mechanisms and the potential molecular targets of H2. Please expand this section to improve your manuscript with relevant study references. Because this one of the possible hypothesis of your manuscript.
Response: we have added “Qin’s group demonstrated that H2 could improve the protein composition and func-tionality of HDL by modulating the oxidized phospholipids (OxPLs)-related enzymes binding on HDL” and “Ohta recently proposed that the oxidative potent of hydroxyl radicals can be alleviated by H2-targeting porphyrin, which contributes to the activation of Nrf2 as a hormesis-like effect”.
- Many references used very old throughout of the manuscript . Please revise more recent references relevant with your study.
Response: we have replace these references with more recent references, such as [22], [90], [108], [109].
- Please check all syntax, typo and grammatical errors throughout of the manuscript.
Response: we have carefully reviewed and revised the syntax, typo and grammatical errors, which marked with red in the revised manuscript.
Round 2
Reviewer 2 Report
Dear authors,
Thanks for improving your manuscript.
Good luck!